# Short-term impacts of Universal Basic Income on population mental health inequalities in the UK: A microsimulation modelling study

Rachel M. Thomson[1]*, Daniel Kopasker[1], Patryk Bronka[2], Matteo Richiardi[2], Vladimir Khodygo[1], Andrew J. Baxter[1], Erik Igelström[1], Anna Pearce[1], Alastair H. Leyland[1], S. Vittal Katikireddi[1]

1 MRC/CSO Social and Public Health Sciences Unit, University of Glasgow, Glasgow, Scotland, United Kingdom, 2 Institute for Social and Economic Research, University of Essex, Essex, England, United Kingdom

* Rachel.Thomson@glasgow.ac.uk

## Abstract

**Data Availability Statement:** UKMOD is freely available for download on the Centre for Microsimulation and Policy Analysis (CeMPA)

### Background

Population mental health in the United Kingdom (UK) has deteriorated, alongside worsening socioeconomic conditions, over the last decade. Policies such as Universal Basic Income (UBI) have been suggested as an alternative economic approach to improve population mental health and reduce health inequalities. UBI may improve mental health (MH), but to our knowledge, no studies have trialled or modelled UBI in whole populations. We aimed to estimate the short-term effects of introducing UBI on mental health in the UK working-age population.

### Methods and findings

Adults aged 25 to 64 years were simulated across a 4-year period from 2022 to 2026 with the SimPaths microsimulation model, which models the effects of UK tax/benefit policies on mental health via income, poverty, and employment transitions. Data from the nationally representative UK Household Longitudinal Study were used to generate the simulated population (*n* = 25,000) and causal effect estimates. Three counterfactual UBI scenarios were modelled from 2023: "Partial" (value equivalent to existing benefits), "Full" (equivalent to the UK Minimum Income Standard), and "Full+" (retaining means-tested benefits for disability, housing, and childcare). Likely common mental disorder (CMD) was measured using the General Health Questionnaire (GHQ-12, score ≥4). Relative and slope indices of inequality were calculated, and outcomes stratified by gender, age, education, and household structure. Simulations were run 1,000 times to generate 95% uncertainty intervals (UIs). Sensitivity analyses relaxed SimPaths assumptions about reduced employment resulting from Full/Full+ UBI.

Partial UBI had little impact on poverty, employment, or mental health. Full UBI scenarios practically eradicated poverty but decreased employment (for Full+ from 78.9% [95% UI 77.9, 79.9] to 74.1% [95% UI 72.6, 75.4]). Full+ UBI increased absolute CMD prevalence by

website: https://www.microsimulation.ac.uk/ukmod/access/. Prospective users must first show their eligibility to utilise the source input data, which is freely available from the UK Data Service provided the user accepts the Terms and Conditions of the End User License (https://www.understandingsociety.ac.uk/documentation/access-data). SimPaths is available open-access on GitHub: https://github.com/centreformicrosimulation/SimPaths. The analytical code in R is available open-access on GitHub: https://github.com/rachelmthomson/thomson-microsim-analysis.

**Funding:** This work was supported by the Wellcome Trust (218105/Z/19/Z [RT] and 205412/Z/16/Z [AP]), European Research Council (949582 [SVK]), Health Foundation (2135162 [SVK]), Medical Research Council (MC_UU_00022/2 [AL]) and Chief Scientist Office (SPHSU17 [AL]). The funders had no role in study design, data collection and analysis, decision to publish, or preparation of the manuscript.

**Competing interests:** The authors have declared that no competing interests exist.

**Abbreviations:** CMD, common mental disorder; COVID-19, Coronavirus Disease 2019; HEED, Health Equity and its Economic Determinants; MIS, Minimum Income Standard; OECD, Organisation for Economic Co-operation and Development; PPI, patient/public involvement; UBI, Universal Basic Income; UI, uncertainty interval; UK, United Kingdom.

0.38% (percentage points; 95% UI 0.13, 0.69) in 2023, equivalent to 157,951 additional CMD cases (95% UI 54,036, 286,805); effects were largest for men (0.63% [95% UI 0.31, 1.01]) and those with children (0.64% [95% UI 0.18, 1.14]). In our sensitivity analysis assuming minimal UBI-related employment impacts, CMD prevalence instead fell by 0.27% (95% UI −0.49, −0.05), a reduction of 112,228 cases (95% UI 20,783, 203,673); effects were largest for women (−0.32% [95% UI −0.65, 0.00]), those without children (−0.40% [95% UI −0.68, −0.15]), and those with least education (−0.42% [95% UI −0.97, 0.15]). There was no effect on educational mental health inequalities in any scenario, and effects waned by 2026.

The main limitations of our methods are the model's short time horizon and focus on pathways from UBI to mental health solely via income, poverty, and employment, as well as the inability to integrate macroeconomic consequences of UBI; future iterations of the model will address these limitations.

## Conclusions

UBI has potential to improve short-term population mental health by reducing poverty, particularly for women, but impacts are highly dependent on whether individuals choose to remain in employment following its introduction. Future research modelling additional causal pathways between UBI and mental health would be beneficial.

## Author summary

### Why was this study done?

- Universal Basic Income (UBI) is a radical social security policy proposal where everyone in a society would receive a regular, unconditional cash payment.

- It has been suggested that UBI might be beneficial for mental health.

- However, there has never been a trial of a true UBI in a high-income country to know what its potential mental health impacts might be.

### What did the researchers do and find?

- We used a policy model to simulate how a UBI might influence mental health for working-age adults in the United Kingdom.

- We found that the effects of UBI were very sensitive to assumptions we made about how people's employment choices might respond to receiving the money—if people choose to stay in work, UBI may have small benefits for population mental health, but if people are more likely to leave work, population mental health may actually worsen.

- The groups most likely to experience positive mental health effects of UBI were women and those with the lowest educational qualifications.

**What do these findings mean?**

- UBI may have potential to improve population mental health in high-income countries, but only if people do not choose to leave work in response to the policy.

- More real-world research is needed to know how people are likely to respond to receiving a UBI in reality.

- The main limitation of our modelling study is that it looks at how UBI would influence mental health only through income and employment, and other pathways might also be important to include in future research.

## Introduction

Many high-income countries are experiencing economic pressures in the aftermath of the Coronavirus Disease 2019 (COVID-19) pandemic and subsequent global energy crisis [1]. There is considerable evidence recessions are linked to deteriorations in mental health, particularly for those of working age [2,3]. There have been growing calls for a radical policy shift to counter the social consequences of such economic fluctuations, as well as the increasing likelihood of wide-scale job losses and insecurity secondary to increasing automation [4–6]. One such proposal is the replacement of existing social security systems with a Universal Basic Income (UBI), which guarantees each individual a regular, unconditional payment designed to meet basic needs, regardless of household income or personal circumstances [7]. Political arguments advocating for UBI originate from both sides of the political spectrum, with the left noting its potential to lift people out of poverty and reduce stigma associated with benefits [8], and the right highlighting reduced bureaucracy and increased personal autonomy [9]. UBI has also been suggested as a potential approach to reducing health inequalities [8], and recent trials of UBI or UBI-like policies have been undertaken or planned in several high-income countries [10,11].

In the United Kingdom (UK), population mental health in working-age adults has deteriorated over the last 15 years against a background of economic crises, austerity policies, and COVID-19 [12–15]. Mental health inequalities have also increased during this period, particularly for younger working-age adults and women [16,17]. Current economic circumstances have potential to worsen this mental health crisis further [1,18,19]. There has been developing interest in the use of UBI-type approaches within the UK's devolved administrations, with the Scottish Government developing detailed plans for a potential UBI pilot [20] (though this did not move forward to implementation) and the Welsh Government recently introducing a small trial of a basic income for individuals leaving the care system [21].

While historically much evaluative research on UBI has tended to focus on its potential impacts on workforce participation, over the last 2 decades there has been a shift to include a broader range of outcomes, including health [22]. Two recent reviews of interventions similar to UBI suggest they may be beneficial for mental health and well-being [23,24]. However, no studies in these reviews evaluated an actual UBI: most were delivered to restricted population groups, some offered one-off rather than regular payments, and most fell considerably short of providing a long-term liveable income [23]. This is perhaps not surprising, as the very high costs and complexities inherent in delivering UBI alongside existing taxation and welfare systems make it challenging to implement meaningful trials, requiring considerable political buy-

in over many years [11]. As a result, to our knowledge, no fully universal UBI policy has ever been trialled in a high-income country. In such situations, policy modelling studies can provide a useful complement to smaller-scale pilots in informing the evidence base for policy-makers [25]. We therefore aimed to use microsimulation to model the implementation of a population-wide UBI in the UK population and to examine its potential impacts on mental health and inequalities in the working-age population.

## Methods

Microsimulation is a modelling approach that simulates individuals with defined characteristics and evolves them over time according to a predetermined set of rules, allowing for exploration of counterfactual scenarios and inequalities between population subgroups [26]. For this analysis, we used 2 models: a static tax-benefit microsimulation model (UKMOD [27]), which we used to calculate the immediate/overnight effects of policies on individual and household incomes; and SimPaths [28], a dynamic, stochastic, discrete-time microsimulation model of economic outcomes, recently modified to estimate the impact of these policy changes on mental health outcomes over time as part of the Health Equity and its Economic Determinants (HEED) project [29]. We summarise the key elements of the model(s) within the paper, with additional technical specifications available in previous publications [27,28] and in S1 Appendix.

### Intervention design

The 3 UBI scenarios modelled in UKMOD (Table 1) were based on recommendations produced for the Scottish Government outlining a UBI trial suitable for the UK context [20]. Two levels of UBI were proposed: 1 low-level (or "Partial") UBI, set at approximately the level of existing welfare benefits, and 1 high-level (or "Full") UBI, set at the Minimum Income Standard (MIS) developed by the Joseph Rowntree Foundation [30]. The former would test the

**Table 1. Baseline and UBI scenarios modelled.**

|  | 1—Baseline | 2—Partial UBI | 3—Full UBI | 4—Full+ UBI |
|---|---|---|---|---|
| **UBI payment per week** | Nil | 0–15 years: £98.84<br>16+ years: £84.89<br>Pensioners*: £196.53 | 0–15 years: £149.69<br>16+ years: £291.62<br>Pensioners: £246.16 | 0–15 years: £149.69<br>16+ years: £291.62<br>Pensioners: £246.16 |
| **Income tax**** | 20% from £12.6k<br>40% from £37.7k<br>45% from £125.1k | 20% from £12.6k<br>45% from £30k<br>60% from £50k | 59% from £15.2k<br>70% from £30k<br>85% from £50k | 59% from £15.2k<br>70% from £30k<br>85% from £50k |
|  | Benefit categories retained (✓) or suspended (×) | | | |
| **Benefit cap**** | ✓ | × | × | × |
| **Child benefit; income support; pension; unemployment benefits** | ✓ | × | × | × |
| **Means-tested benefits for caring, childcare, disability, housing, and limited capability for work** | ✓ | ✓ | × | ✓ |
| **Maternity pay, statutory sick pay, student benefits** | ✓ | ✓ | ✓ | ✓ |

Baseline = planned tax/benefit policies for UK. Partial UBI = UBI set at the level of existing benefits. Full UBI = UBI set at the level of MIS. Full+ UBI = MIS plus means-tested benefits for caring, childcare, disability, housing, and limited capability for work.

*The pension-age benefit is applied when an individual reaches pensionable age; this differs by gender and with other circumstances in the UK, but is typically 66.

**While there is some minor variation in the lower/upper income tax rates and bands in Scotland, for simplicity, the rates applied in the majority of the UK are shown in the table. In the UBI scenario, the same tax rates were applied to all 4 devolved nations.

***The "benefit cap" (which limits the amount of total benefits that can be delivered to a single household/benefit unit in the UK) was suspended in all UBI scenarios to avoid interference with the UBI delivery.

MIS, Minimum Income Standard; UBI, Universal Basic Income; UK, United Kingdom.

effect of universality and allocation of payments directly to individuals (rather than households), and the latter would test the additional effect of making this payment sufficient to live on. In both cases, it was recommended some current means-tested UK benefits were retained to avoid disadvantaging those with higher needs (Table 1). However, this removes 1 key advantage of UBI purported by advocates—the removal of means-testing and reduction in administrative burden [9]. Therefore, we modelled 2 "Full" UBI scenarios (Table 1): without ("Full," Scenario 3) and with ("Full+," Scenario 4) these means-tested benefits. Full details of individual benefits retained or suspended in each scenario are in the Appendix (Table C in S1 Appendix).

UBI implementation in UKMOD followed the approach of De Henau and colleagues [31]. In Scenario 2 (Partial UBI), income tax rates were increased to around the highest acceptable levels reported in UK surveys of public attitudes to taxation [32,33]. In Scenarios 3 and 4 (Full/Full+ UBI), rates were set at those suggested by Kumar and colleagues when designing a Scottish UBI to meet the 2019 MIS [34]. Income tax thresholds were lowered in all intervention scenarios to reduce the government deficit caused by the policy, though based on patient/public involvement (PPI; see below), we did not try to achieve full fiscal neutrality. The UBI was treated as taxable income, with the Personal Allowance (the amount of tax-free income individuals are allowed in the UK, currently £12,570) set to the level of the UBI payment in Scenarios 3 and 4.

## Model structure

SimPaths evolves a representative sample of the UK population in 1-year increments through 6 modules: Demography, Education, Health, Household composition, Nonlabour market income, and Labour supply (Fig 1). It was developed using JAS-mine, an open-source, object-oriented Java-based platform specifically designed for discrete-event simulations [35]. An open-source version of the model is available on GitHub (https://github.com/centreformicrosimulation/SimPaths). Each module includes several subprocesses estimated on longitudinal data external to the simulation [28]. The input data are described below. Alignment within the ageing and fertility processes ensures simulated population demographics do not deviate substantially from official projections from the Office for National Statistics.

In the Labour supply module, individuals within households (which may be made up of multiple benefit units) select the number of hours they wish to work based on a structural random utility model, with potential wages calculated using a Heckman-corrected wage equation. This is a nonforward-looking model of labour supply, in which employment decisions are made in order to maximise within-period utility of benefit units, given exogenous hourly wage rates predicted by the wage model. The model is unitary, which means that decision-making is at the level of the benefit unit—for example, one single choice involving labour supply decisions for each partner is made by couples, in order to maximise their joint utility.

A proof-of-principle causally informed mental health module was added for HEED (Fig 1, lower panel), estimating an individual's likelihood of experiencing a common mental disorder (CMD) based on their demographic characteristics (Step 1) and on empirical epidemiological estimates of the effects of economic transitions on mental health (Step 2). These economic determinants of health are thought to be important in explaining the development and persistence of health inequalities over time [36,37], but considerable potential for bias, confounding, and reverse causation means application of cross-sectional relationships to predict the effect of counterfactual scenarios may be problematic [29,38]. We therefore estimated the short-term (1-year) effects of income changes and transitions in or out of poverty/employment on mental health in working-age adults, and the short-term effect of spending 2 consecutive years in poverty/unemployment on mental health [38,39].

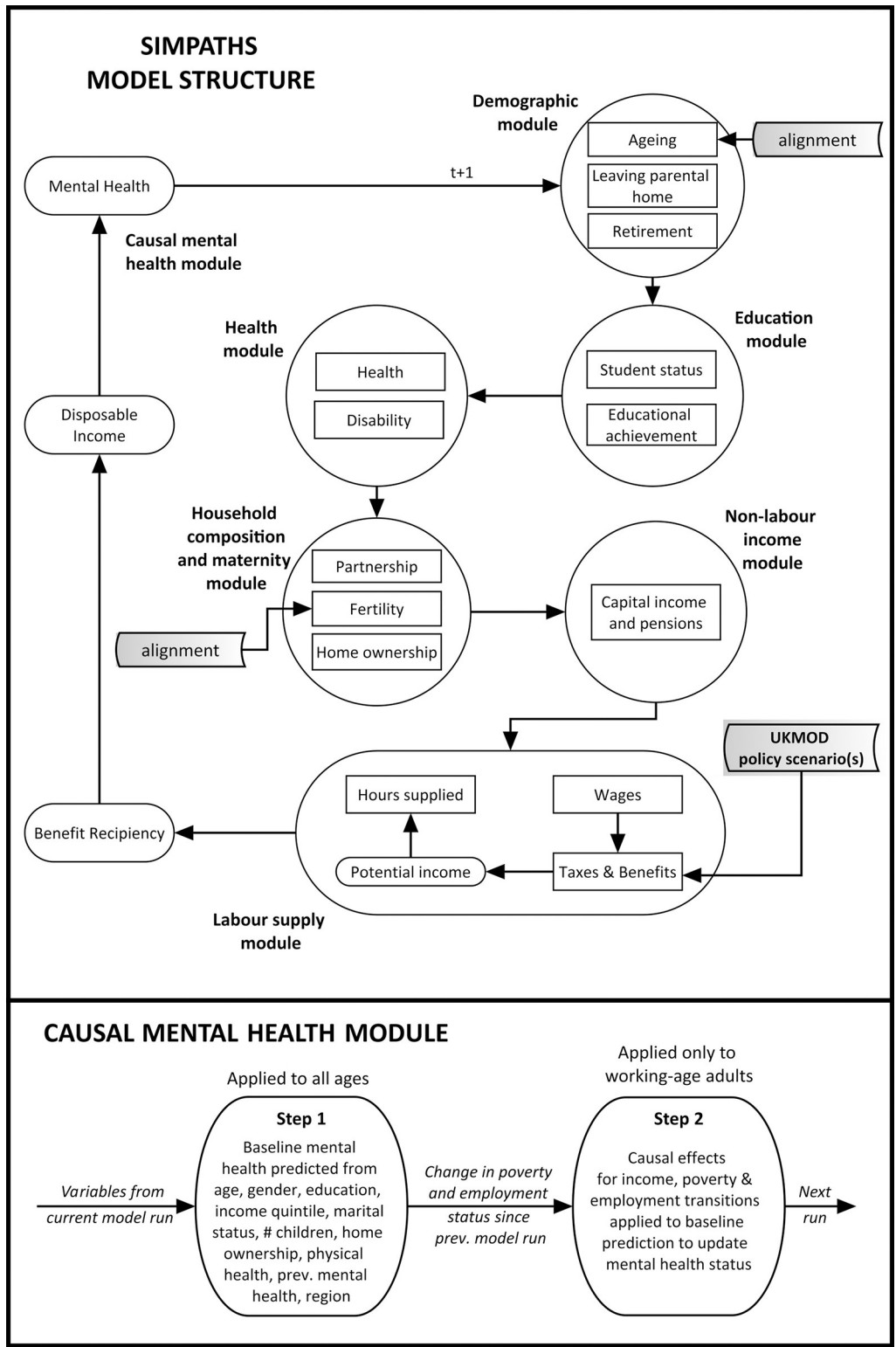

**Fig 1. Structure of the SimPaths model.** Top panel = full SimPaths model structure; bottom panel = additional detail on causal mental health module.

The key structural and theoretical assumptions of SimPaths are illustrated in Table A in S1 Appendix. For this analysis, the strongest assumptions were that any macroeconomic impacts of the policy intervention (which are not modelled within SimPaths) would not influence outcomes and that individuals would respond to UBI income in the same way as any other additional income when deciding whether to work. As it has been suggested that employment impacts of radical UBI-like interventions may be smaller than expected [23,40], we explored the potential influence of this second assumption on our primary analysis (where the model endogenously produces labour supply responses) in 2 structural sensitivity analyses (see below).

Models were subjected to both internal and external validation, as described in S1 Appendix and elsewhere [27,28]. Simulations were run from 2022 to 2026, with UBI introduced from 2023 in intervention scenarios.

## Model input data

UKMOD analyses income data from the Family Resources Survey, an annual representative cross-sectional survey of UK private households [41]. The model generates multiple output files simulating the effect of each annual tax-benefit policy scenario on incomes for every individual within the dataset.

SimPaths' simulated UK population and the parameter estimates for the modules are drawn from the UK Household Longitudinal Study (also called "Understanding Society"), a representative panel-based study that includes income, health, and sociodemographic measures [42]. The labour supply module uses utility functions estimated on the data from the Family Resources Survey [41]—these estimates inform the decisions individuals and benefit units take about how to prioritise labour time over leisure time.

UKMOD output files are integrated into SimPaths' simulation processes in the Labour Supply module (Fig 1). Households in SimPaths are probabilistically matched with "donor" households from the relevant UKMOD output file with similar characteristics, with this matched unit then used to convert the SimPaths household's simulated gross income (including earnings, capital income, and pensions) to disposable income (i.e., subtracting taxes and adding benefits) according to the selected tax-benefit regime for that year.

The mental health module also uses data from Understanding Society (Table B in S1 Appendix). The Step 2 causal estimates for transitions between poverty and employment states in the working-age population aged 25 to 64 years were calculated using double-robust marginal structural modelling; we draw on our previous causal epidemiological analyses (with full methodological details including our causal framework and estimation approach published previously [38,39]). As effects differed by gender (but not markedly by education level or age group), estimates in the mental health module were calculated separately for men and women.

## Model outputs

Output produced by the model consists of an SQL database tables and/or CSV files at the individual, benefit unit, and household levels, which can be linked through the unique identifiers. The output files contain the values of simulated variables for each individual unit in each year of the simulation, effectively producing a "synthetic" panel dataset. The initial UKMOD output also included a calculation of the Gini coefficient for each policy scenario, a commonly used measure of income inequality from 0 (perfect equality) to 1 (perfect inequality), which expresses the expected absolute gap between people's incomes relative to the mean population income [43].

We calculated all outcomes in the working-age population aged 25 to 64 years, comparing trends in median income, poverty, employment, and mental health for baseline versus

intervention scenarios. Employment was defined as being in any form of paid work, including self-employment. Poverty was defined as an annual income less than 60% of the median in the baseline policy scenario, before housing costs.

Our primary health outcome was the binary version of the General Health Questionnaire (GHQ-12), which identifies symptoms of psychological distress [44]. Scoring 4 or more indicates a likely CMD. CMD prevalence was calculated for the whole working-age population and for population subgroups of interest: gender (male versus female), education (high versus medium versus low), age (25 to 44 years versus 45 to 64 years), and household structure (has children versus no children; if lone parent). Number of additional/reduced CMD cases was calculated using Organisation for Economic Co-operation and Development (OECD) estimates of the size of the UK working-age population in Quarter 4 of 2022 (41.6 million) [45]. Our secondary outcome was the 36-point GHQ Likert score, a continuous measure scored from the same questionnaire, with higher scores indicating higher levels of psychological distress.

Relative and slope indices of inequality (RII/SII) in CMD (by education) were calculated for the whole working-age population, and for men and women separately. These inequality measures regress the outcome for those in a particular socioeconomic group on the proportion of the population that has a higher position in the hierarchy, in this case, those with higher education [46]. They can be interpreted as the ratio (RII) or absolute difference (SII) in CMD prevalence between those with the hypothetically least and the most education, taking into account population size.

## Sensitivity analyses

To investigate assumptions about individuals' decision-making and experiences of employment following receipt of a liveable UBI, we conducted 2 structural sensitivity analyses for Scenarios 3 and 4 (Full and Full+ UBI). Firstly, to simulate the possibility there would be no marked fall in employment in response to UBI delivery, we modified utility values in the SimPaths Labour Supply module so employment rates remained constant between the baseline and intervention scenarios. Secondly, to simulate the possibility that moving out of employment following UBI receipt may be more akin to the effect of voluntarily exiting the labour force, we substituted the causal effect estimates for employment transitions used in Step 2 of the mental health module with those for economic inactivity for our primary outcome (Table D in S1 Appendix).

Finally, an additional analytical sensitivity analysis investigated whether the patterning of findings differed when using effect estimates from systematic reviews for our primary outcome (Table D in S1 Appendix).

## Uncertainty analysis

SimPaths accounts for parameter uncertainty by including routines that facilitate bootstrapping parameter estimates, based on estimated point values and covariance matrices. This involves repeated simulations, each based on a different random draw for model parameters. Similarly, Montecarlo variation can be explored by conducting repeated simulations each based on fresh set of random draws or by arbitrarily scaling-up the simulated population size. These methods can be used to generate a distribution of model outcomes, around central projections.

To account for stochastic and parameter uncertainty, we ran 1,000 simulations for each analysis, with parameter estimates for all processes randomly drawn from a distribution based on their variance. Results are presented as the median of the outcomes from all 1,000 simulations, and the difference in medians between the intervention scenario and baseline, with 95%

uncertainty intervals (UIs) generated from the 2.5th and 97.5th percentiles. Data were analysed in RStudio version 2022.12.0+353.

### Patient and public involvement

Our Advisory Group included representation from third sector organisations including the Mental Health Foundation, Joseph Rowntree Foundation, and Basic Income Conversation, and policymakers from Public Health Scotland and Public Health Wales. The group informed all preparatory work on causal modelling [38,39] and shaped the selection and development of UBI scenarios (see S1 Appendix). As there is considerable debate on the optimal way to fund a UBI, and many proposals include the use of novel tax levers that are difficult to model (e.g., wealth or carbon taxes; [47]), the group felt strongly that we should focus explicitly on health impacts rather than aiming for fiscal neutrality.

## Results

### Model performance

Internal validation of SimPaths, including parameter sweeps and stress-testing with extreme scenarios, suggested the model performed as expected, albeit with small overestimates of employment rates and earnings for those with low education (Figs A and B in S1 Appendix). On external validation, our predicted CMD prevalence for 2012 to 2018 was broadly comparable with observed data from the Health Survey for England [48], though trends were flatter (Fig C in S1 Appendix).

### Fiscal and distributive impacts of UBI

All 3 UBI scenarios modelled in UKMOD increased disposable incomes more for those with lower starting incomes, reducing income inequality: the Gini coefficient reduced by 0.04 for the Partial UBI scenario, 0.15 for Full UBI, and 0.16 for Full+ UBI (Table E and Figs D-F in S1 Appendix). Partial UBI resulted in a fiscal deficit of £95.6 billion (bn); deficits for the Full/Full+ UBI scenarios were £30.9bn and £65.2bn, respectively.

### Primary analysis

In the primary analysis, levels of poverty were reduced in all UBI scenarios compared with Baseline from the year of policy implementation, with this reduction being sustained throughout the rest of the simulation (Fig 2). The reduction was considerably more marked for the Full UBI scenarios, where poverty reduced to 0.01% (95% UI 0.00, 0.03) for Full+ UBI in 2023, compared with 7.13% (95% UI 6.62, 7.66) in the Partial UBI scenario, and 9.10% (95% UI 8.47, 9.71) at Baseline (Table F in S1 Appendix). Median annual incomes in 2023 were higher in all UBI scenarios compared with the Baseline of £22,578 (95% UI 22,111, 23,015): £27,173 (95% UI 26,599, 27,653) with Partial UBI, £27,421 (95% UI 27,191, 27,630) with Full UBI, and £27,719 (95% UI 27,474, 27,922) with Full+ UBI (Table F in S1 Appendix).

For employment rates, there was a small increase from Baseline in the Partial UBI scenario (Fig 2), from 78.93% (95% UI 77.94, 79.86) to 79.38% (95% UI 78.41, 80.24), though this waned by 2026. Employment fell in Full UBI scenarios, to 75.69% (95% UI 74.44, 76.92) with Full and 74.10% (95% UI 72.62, 75.43) with Full+ UBI (Table F in S1 Appendix). Mean weekly hours worked reduced by 1.76 hours (95% UI −2.10, −1.42) with Full and 2.31 hours (95% UI −2.72, −1.91) with Full+ UBI (Table F in S1 Appendix). As with poverty, the differential trajectories of lower employment for the Full UBI policies compared with Baseline were sustained throughout the study period following implementation.

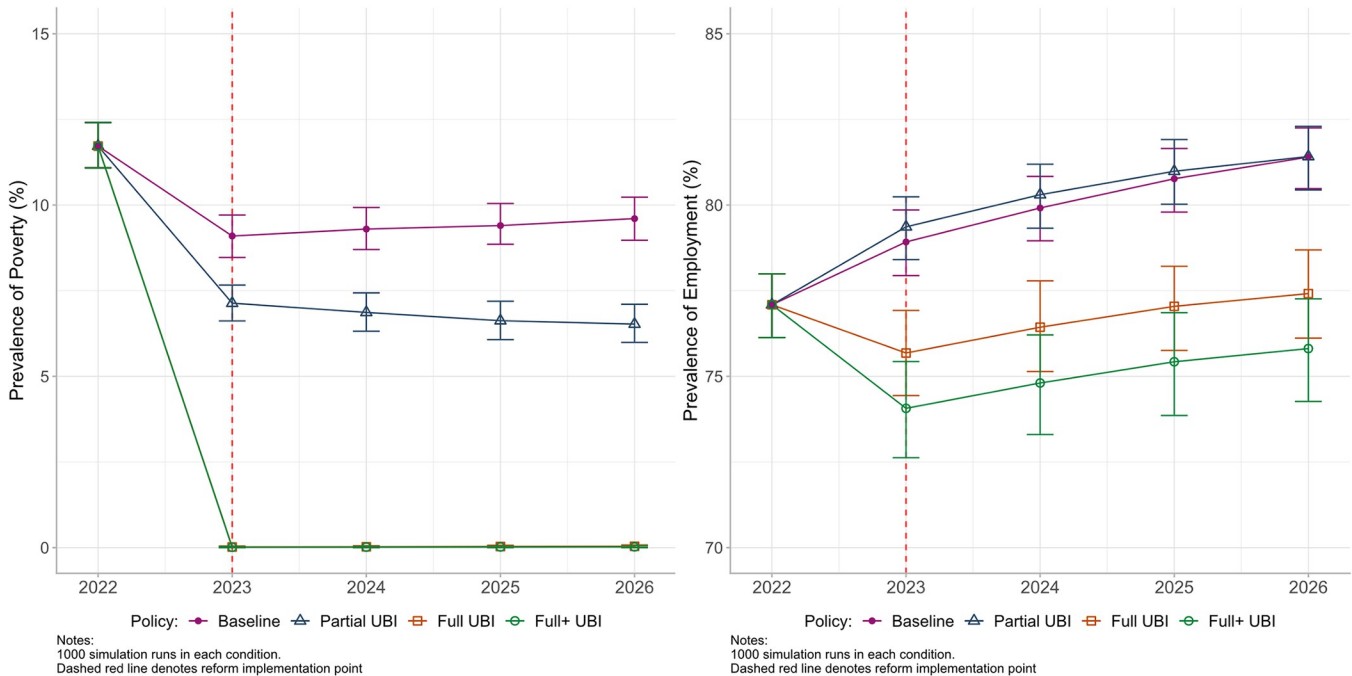

**Fig 2. Estimates of poverty (left panel) and employment (right panel) for modelled UBI scenarios from 2022 to 2026.** Baseline = planned tax/benefit policies for UK. Partial UBI = UBI set at the level of existing benefits. Full UBI = UBI set at the level of MIS. Full+ UBI = MIS plus means-tested benefits for caring, childcare, disability, housing, and limited capability for work. Whiskers = 95% UIs. Note that in the left panel, Full and Full+ UBI lines overlap around zero from 2023 onwards. MIS, Minimum Income Standard; UBI, Universal Basic Income; UI, uncertainty interval; UK, United Kingdom.

In the Partial UBI scenario, resultant changes in income, poverty, and employment led to a CMD prevalence which did not differ markedly from Baseline throughout the study period (Fig 3). In the Full UBI scenarios, CMD prevalence was slightly higher than Baseline in the first 2 years of implementation, though this effect was small and waned over time: the absolute difference in prevalence for Full+ UBI was 0.38% (95% UI 0.13, 0.69) in 2023 and 0.08% (95% UI -0.78, 0.94) in 2026 (Table G in S1 Appendix). In 2023, this would equate to approximately 157,951 additional CMD cases (95% UI 54,036, 286,805). There was no notable influence of any UBI scenario on inequalities in CMD prevalence by education using either the RII or SII; for example, with Full+ UBI, the RII in 2023 reduced by 0.03 points (95% UI −0.09, 0.02) from 1.33 (95% UI 1.13, 1.56) at Baseline (Fig G and Table G in S1 Appendix).

On stratification by gender, education, age, and household structure (Fig 4), the short-term worsening of mental health with Full+ UBI remained small, albeit slightly more pronounced for men (0.63% [95% UI 0.31, 1.01]), those with high education (0.48% [95% UI 0.13, 0.91]), and those with children (0.64% [95% UI 0.18, 1.14]) (Table H in S1 Appendix). For men only, this worsening was potentially sustained over time with a difference of 0.29% (95% UI −0.90, 1.41) remaining in 2026 compared to Baseline, equivalent to an additional 60,018 men with CMD (95% UI −186,264, 291,814); however, UIs were very wide. The only population subgroups who potentially benefited from the implementation of Full UBI by 2026 were women (−0.15% [95% UI −1.53, 1.24]) and lone parents (−0.23% [95% UI −6.82, 5.91]), and, again, UIs were wide.

## Sensitivity analyses

The results of all sensitivity analyses are summarised in Table 2 (shown in full in Tables I-M and Figs H-P in S1 Appendix). Our first structural sensitivity analysis modelled the possibility

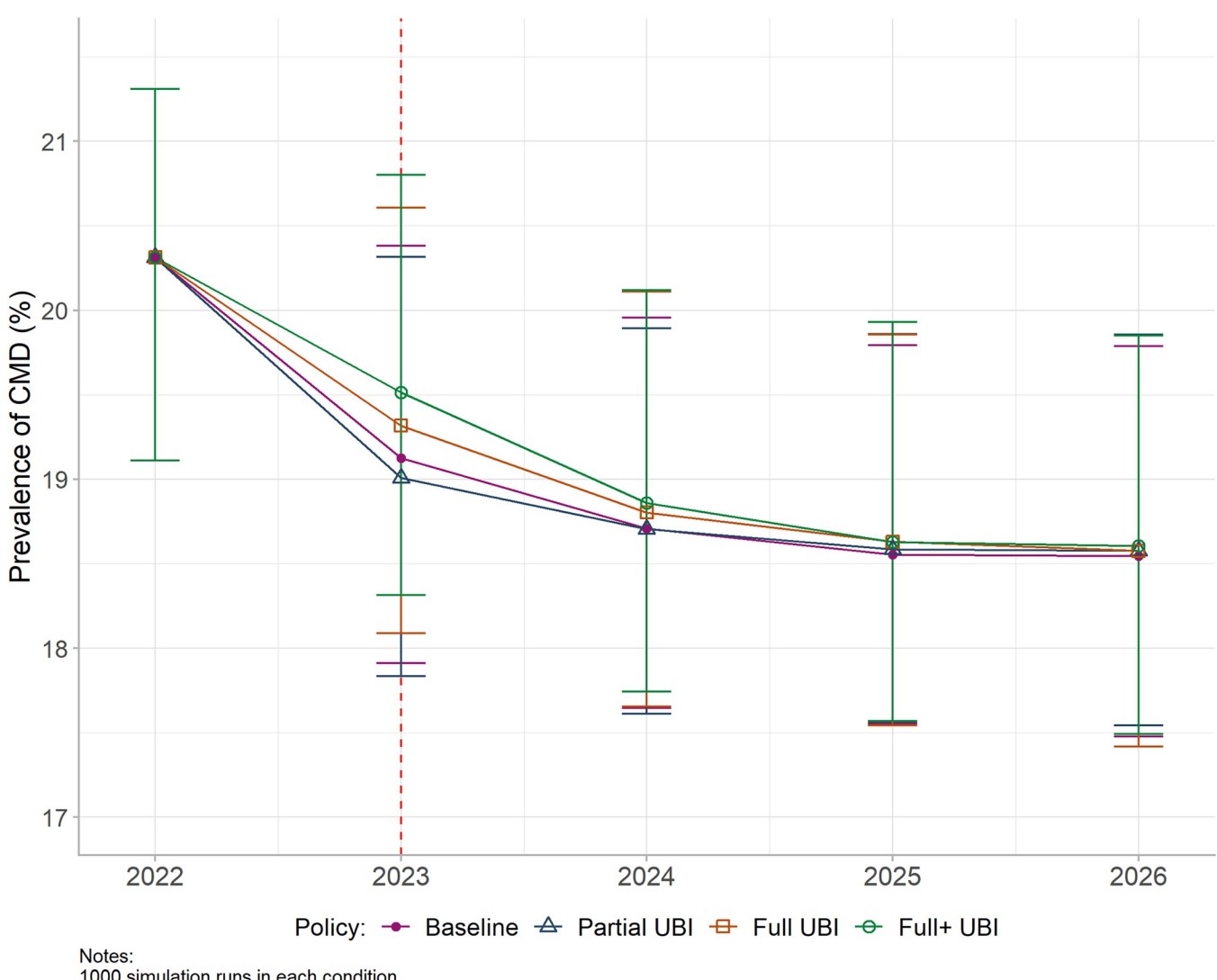

**Fig 3. Estimated prevalence of CMD for modelled UBI policies from 2022 to 2026.** Baseline = planned tax/benefit policies for UK. Partial UBI = UBI set at the level of existing benefits. Full UBI = UBI set at the level of MIS. Full+ UBI = MIS plus means-tested benefits for caring, childcare, disability, housing, and limited capability for work. Whiskers = 95% UIs. CMD, common mental disorder; MIS, Minimum Income Standard; UBI, Universal Basic Income; UI, uncertainty interval; UK, United Kingdom.

people would be less likely to exit employment in response to the most generous UBI scenarios. Here, poverty reductions associated with both Full and Full+ UBI interventions remained the same, but employment changes were minimised to approximate the Baseline scenario (Fig H and Table I in S1 Appendix). Under these conditions, in a reversal of the primary analysis findings, there was a small short-term reduction in prevalence of CMD in the UBI scenarios: −0.27% (95% UI −0.49, −0.05) in 2023 for Full+ UBI, equivalent to a reduction of 112,228 CMD cases (95% UI 20,783, 203,673) (Table 2). As in the primary analysis, these differences waned over time, and there was little impact on inequality measures. On stratification, the groups for whom the Full+ UBI brought most short-term improvement in CMD prevalence were women (−0.32% [95% UI −0.65, 0.00] in 2023), those with low education (−0.42% [95% UI −0.97, 0.15]), and those without children (−0.40% [95% UI −0.68, −0.15]), though these improvements were not sustained past 2023 (Fig M and Table K in S1 Appendix).

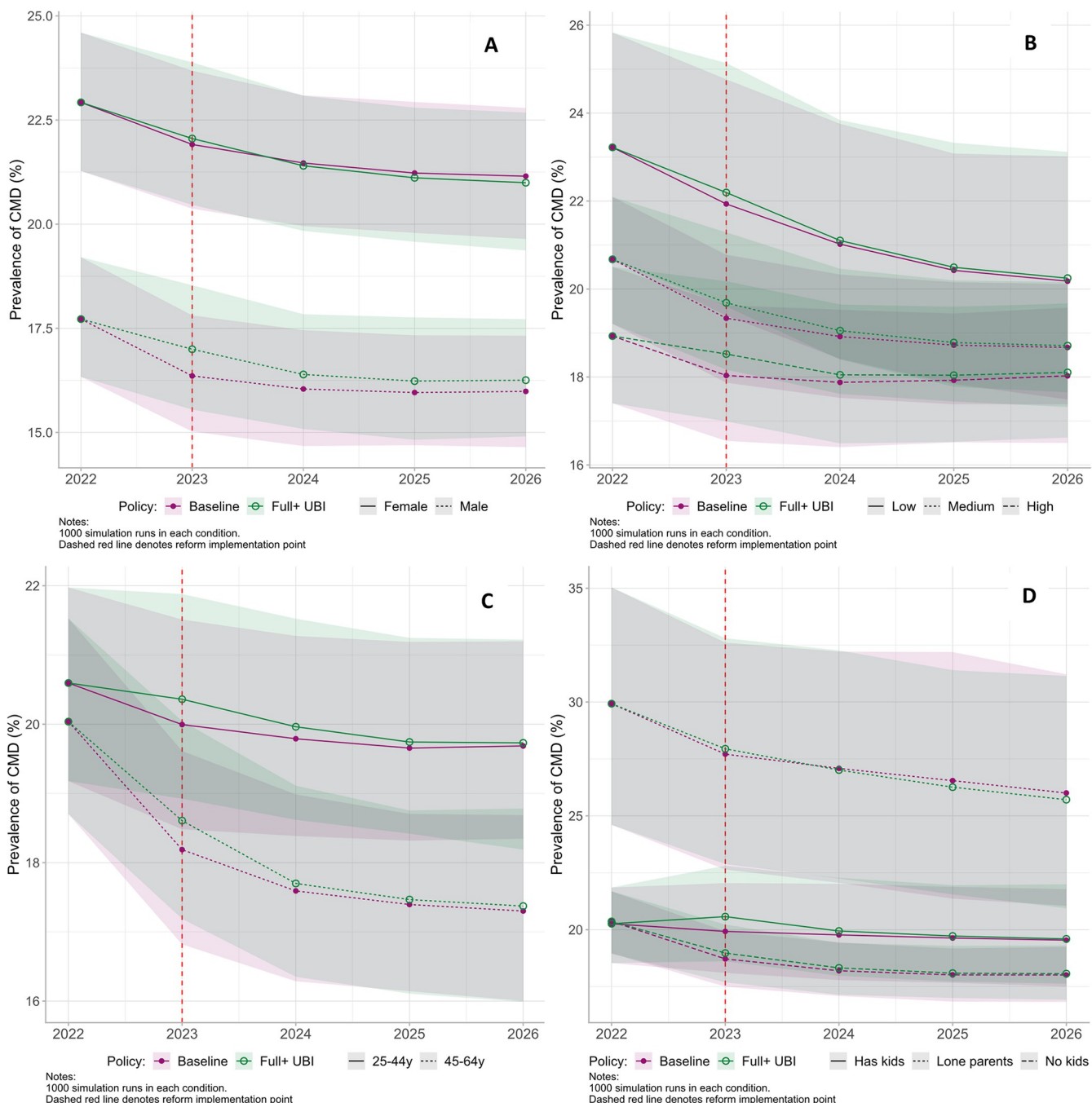

**Fig 4. Estimated prevalence of CMD for modelled UBI policies from 2022 to 2026 with 95% UIs, stratified by gender (A), education (B), age (C), and household structure (D). Note different scales used for each stratification.** y = years. Baseline = planned tax/benefit policies for UK. Full+ UBI = UBI set at the MIS plus means-tested benefits for caring, childcare, disability, housing, and limited capability for work. Ribbons = 95% UIs. Low education = no formal qualifications; medium education = Higher/A-level/GCSE or equivalent; high education = degree or equivalent. CMD, common mental disorder; GCSE, General Certificate of Secondary Education; MIS, Minimum Income Standard; UBI, Universal Basic Income; UI, uncertainty interval; UK, United Kingdom.

A second structural sensitivity analysis modelled the possibility that voluntary transitions out of employment under a Full/Full+ UBI would have a less negative impact on mental health than standard unemployment transitions. Under these conditions, CMD prevalence with Full + UBI was again lower than Baseline in 2023 by −0.31% (95% UI −0.93, 0.35) and waned

**Table 2. Difference in prevalence of CMD and CMD inequality measures for Full+ UBI policies versus Baseline policies from 2023 to 2026.**

| ANALYSIS | 2023 | 2024 | 2025 | 2026 |
|---|---|---|---|---|
| | **ABSOLUTE % DIFFERENCE IN CMD PREVALENCE** | | | |
| **1. Main analysis** | 0.38% (0.13, 0.69) | 0.16% (−0.78, 0.94) | 0.08% (−0.73, 0.89) | 0.08% (−0.78, 0.94) |
| *N. of CMD cases* | *+157,951 cases (54,036, 286,805)* | *+66,506 cases (−324,215, 390,720)* | *+33,253 cases (−303,432, 369,937)* | *+33,253 cases (−324,215, 390,720)* |
| **2. Reduced unemployment** | −0.27% (−0.49, −0.05) | −0.11% (−0.88, 0.69) | −0.02% (−0.88, 0.72) | −0.06% (−0.95, 0.76) |
| *N. of CMD cases* | *−112,228 cases (−203,673, −20,783)* | *−45,723 cases (−365,781, 286,805)* | *−8,313 cases (−365,781, 299,275)* | *−24,940 cases (−394,877, 315,902)* |
| **3. Economic inactivity effects** | −0.31% (−0.93, 0.35) | −0.16% (−1.23, 0.82) | −0.13% (−1.23, 1.03) | −0.11% (−1.33, 1.11) |
| *N. of CMD cases* | *−128,855 cases (−386,564, 145,481)* | *−66,506 cases (−511,262, 340,841)* | *−54,035 cases (−511,262, 428,130)* | *−45,723 cases (−552,828, 461,383)* |
| **4. Alternative SR causal estimates** | 0.05% (−0.14, 0.22) | 0.04% (−0.74, 0.84) | 0.03% (−0.82, 0.88) | 0.04% (−0.80, 0.87) |
| *N. of CMD cases* | *+20,783 cases (−58,192, 91,445)* | *+16,626 cases (−307,588, 349,154)* | *+12,470 cases (−340,841, 365,781)* | *+16,626 cases (−332,528, 361,624)* |
| | **DIFFERENCE IN RELATIVE INDEX OF INEQUALITY (RII)** | | | |
| **1. Main analysis** | −0.03 (−0.09, 0.02) | −0.01 (−0.26, 0.22) | −0.01 (−0.27, 0.24) | −0.01 (−0.25, 0.24) |
| **2. Reduced unemployment** | −0.02 (−0.06, 0.03) | −0.01 (−0.25, 0.23) | −0.00 (−0.26, 0.25) | −0.01 (−0.24, 0.23) |
| **3. Economic inactivity effects** | −0.01 (−0.09, 0.07) | −0.00 (−0.24, 0.24) | −0.00 (−0.25, 0.25) | 0.00 (−0.24, 0.24) |
| **4. Alternative SR causal estimates** | −0.02 (−0.06, 0.02) | −0.01 (−0.27, 0.24) | −0.00 (−0.25, 0.25) | −0.01 (−0.25, 0.25) |
| | **DIFFERENCE IN SLOPE INDEX OF INEQUALITY (SII)** | | | |
| **1. Main analysis** | −0.00 (−0.01, 0.00) | −0.00 (−0.04, 0.03) | −0.00 (−0.04, 0.04) | −0.00 (−0.04, 0.04) |
| **2. Reduced unemployment** | −0.00 (−0.01, 0.00) | −0.00 (−0.04, 0.03) | −0.00 (−0.04, 0.04) | −0.00 (−0.04, 0.03) |
| **3. Economic inactivity effects** | −0.00 (−0.01, 0.01) | −0.00 (−0.03, 0.03) | −0.00 (−0.04, 0.04) | −0.00 (−0.04, 0.04) |
| **4. Alternative SR causal estimates** | −0.00 (−0.01, 0.00) | −0.00 (−0.04, 0.03) | −0.00 (−0.04, 0.03) | −0.00 (−0.04, 0.04) |

SR = systematic review; N. = number. Baseline = planned tax/benefit policies for UK. Partial UBI = UBI set at the level of existing benefits. Full UBI = UBI set at the level of MIS. Full+ UBI = MIS plus means-tested benefits for caring, childcare, disability, housing, and limited capability for work. 95% UIs in brackets. Analysis 1 is the primary analysis indicating maximal employment effects of UBI. Analysis 2 altered participant decision-making processes to approximate a scenario with minimal employment effects of UBI, i.e., where fewer people gave up work and/or opted to stay out of employment as a result of the income increase. Analysis 3 maintained the employment changes from Analysis 1, but treated moves out of employment as representing moves into economic inactivity (which are less detrimental to mental health). Analysis 4 used causal effect estimates for poverty and employment transitions sourced from systematic reviews, rather than our epidemiological analyses. Absolute number of CMD cases calculated using OECD estimates of the size of the UK working age population in Quarter 4 of 2022 (41,566,000).

CMD, common mental disorder; MIS, Minimum Income Standard; OECD, Organisation for Economic Co-operation and Development; UBI, Universal Basic Income; UI, uncertainty interval; UK, United Kingdom.

slightly less over the study period, but there were very wide UIs for all estimates (Table 2). On stratification, in contrast with other analyses, this was driven mostly by improvements for men, with a −0.58% (95% UI −1.52, 0.37) reduction in male CMD prevalence with Full+ UBI in 2023, and a −0.61% (95% UI −2.32, 1.06) reduction in 2026 (Fig N and Table K in S1 Appendix). Other stratified results were patterned similarly to the primary analysis.

In our analytical sensitivity analysis using alternative causal estimates from systematic reviews, effect magnitudes were smaller, but we found no marked divergence from our primary findings, i.e., Partial UBI generated small improvements to poverty but no clear mental health improvement, and Full/Full+ UBI scenarios generated large improvements to poverty, reductions in employment, and slightly higher CMD prevalence for short periods of time (Figs O and P and Tables L and M in S1 Appendix).

Results for our secondary outcome (GHQ Likert score, which represents a continuous measure of psychological distress rather than a binary measure of likelihood of CMD) are shown in Tables N and O in S1 Appendix. These were in keeping with the findings for our primary outcome, with results from the main analysis showing higher GHQ Likert scores with Full + UBI compared to the baseline scenario (0.03 [95% UI 0.00, 0.06]) and the structural sensitivity analysis with reduced employment assumptions showing instead a reduced GHQ score (−0.06 [95% UI −0.08, −0.04]). On stratification, as with the primary outcome, the positive

effects of UBI in this sensitivity analysis were larger for women (−0.08 [95% UI −0.12, −0.05]) and those with least education (−0.09 [95% UI −0.13, −0.06]), and there were also larger effects for lone parents (−0.08 [95% UI −0.15, −0.01]).

## Discussion

Our findings suggest a low UBI set at the level of existing welfare benefits is unlikely to markedly affect population mental health in the UK, despite considerable financial cost. For a UBI set at the level of a liveable income, impacts are less clear and highly dependent on associated employment effects. If people are less likely to work, mental health may deteriorate in the short term (particularly for men), whereas if employment levels remain unchanged, a small improvement in mental health is anticipated, with women and those with low education seeing most improvement. For a comprehensive Full+ UBI, our analyses suggest a worst-case scenario of 157,951 additional cases of common mental disorders (95% UI 54,036, 286,805) if employment rates fall, and a best-case scenario of 112,228 fewer cases (95% UI 20,783, 203,673) if they do not. In both situations, impacts appear relatively short-lived, with no meaningful impact on mental health inequalities by education. There is also considerable uncertainty around many of our estimates, particularly after the first year of policy implementation.

These findings are driven by the relative importance of income, poverty, and employment for mental health [49]. While there is a well-evidenced and large cross-sectional association between income and health, evidence for a longitudinal or causal effect of income has been less clear [50]. Our recent systematic review found that, while income changes do impact future mental health and well-being, the effect size is perhaps smaller than anticipated—though moves across a poverty threshold exert the largest of these small effects [51]. Contrastingly, several systematic reviews report large effects of employment transitions on mental health [52,53]. In keeping with this review evidence, the causal effects we estimated for poverty transitions were around 11% of the size of those for employment transitions (Table B in S1 Appendix). This explains why the modelled mental health effects of a policy with such considerable impacts on poverty levels are smaller than might be anticipated, and why findings are so sensitive to assumptions regarding employment.

It is not unreasonable to assume labour supply effects of UBI may be minimal, making our primary analysis a reasonable "worst-case scenario" for mental health. There is little evidence UBI-like interventions are associated with large increases in unemployment [23], with existing reviews finding few and surprisingly small adverse labour supply effects [40,54]. This is in keeping with more recent real-world evidence from the introduction of COVID-related benefits, which were found to result in minimal work disincentives in the United States despite considerable expansion compared with the prepandemic provision of welfare [55,56]. On UBI's potential health effects, 2 recent reviews report consistent evidence of improved mental health and well-being for UBI-like policies in trials, more so than other adult health outcomes [23,24]. In contrast, we report here very modest and uncertain findings even for our "best-case scenario."

This could be because we estimate only material pathways from UBI to mental health via income, poverty, and employment, rather than psychosocial pathways of perceived social/economic status or security. There is some evidence UBI partially improves well-being through increased confidence, financial independence, and stability [57,58]. Our model will not capture these mechanisms if they do not act solely through income levels or employment status, potentially underestimating some positive effects of UBI on mental health [59]. In our future work, we intend to expand the included causal pathways to reduce this limitation. A recent microsimulation study modelling UBI and mental health in children and young people also

reported larger effects than we do [60,61], though this is not necessarily surprising as it is known income interventions are more likely to improve health in children than adults [62,63], and employment transitions are less relevant to this population group. Future modelling work combining these 2 population groups would be of interest and beneficial in determining to what extent our estimates may be conservative.

As the first microsimulation study, to our knowledge, considering the adult health impacts of a population-wide UBI, our work has many important strengths. We modelled a range of policy scenarios and sensitivity analyses developed through third sector and policymaker engagement. We use 2 established microsimulation models and draw on advanced epidemiological methods to estimate effects for parameterisation, with our causal framework and estimation process clearly described and previously published [38,39]. We present a range of validation results, demonstrating that this model performs satisfactorily. Finally, we report results across a broad range of population subgroups to interrogate differential effects, as well as assessing effects on measures of mental health inequalities. There are, however, some limitations to our work. We model only short-term (1-year) causal effects of economic transitions and deal with a relatively short time horizon (4 years from implementation), potentially underestimating longer-term impacts of the policies and introducing the more marked uncertainty seen around the estimates in the years after the intervention. We cannot model interactions between our simulated population and the macroeconomy, and in reality, changes to macroeconomic factors such as economic growth, public service provision, and income inequality are highly likely under UBI [20]. Adding these interactions via integration with macroeconomic models is complex [64], but we intend to explore this possibility in future work to overcome this limitation. Finally, by including only the causal effects of income, poverty, and employment transitions, we assume these are the only meaningful causal pathways through which a UBI will influence population mental health, which may not be the case [23,24].

In terms of policy implications, even with changes to income tax rates far beyond what is considered publicly acceptable [32,33], the fiscal deficit associated with our most comprehensive UBI is £65.2bn. In our best-case scenario for mental health outcomes, this equates to a cost of just under £600,000 per case of CMD avoided, substantially higher than the approximately £350/month individuals state they would be willing to pay to avoid depression [65], which might suggest such a policy approach should not be pursued based on its mental health benefits alone. However, we do note that this does not take into account potential improvements in child and adolescent mental health, which may be larger and longer-lasting [61]. Given our principal finding that mental health effects of UBI may be contingent on changes in employment rates, policymakers considering this approach should proactively consider how to mitigate or prevent such changes and may wish to pilot their chosen policy with close monitoring of employment outcomes prior to broader implementation. Trial data could be usefully compared to outputs from this modelling to determine which labour market scenario is most likely, informing future policy modelling and implementation decisions. Future research addressing the limitations of our modelling would be useful, in particular to expand the causal pathways considered, integrate the microsimulation with macroeconomic simulations, and include longer-term economic exposures over a longer time period [29]. Finally, using "real-world" data from ongoing UBI trials (such as the Welsh care leavers study [21]) to further refine the labour supply elements of the model or generate improved causal effect estimates would be highly valuable.

In conclusion, while it comes with considerable financial cost, our exploratory modelling analyses suggest a liveable UBI may reduce the number of UK working-age adults diagnosed with a common mental health problem by around 112,000 cases on introduction, if recipients choose to remain in work. However, in the worst-case scenario for employment effects, our

simulations suggest the same policy could instead lead to an increase of 157,951 cases. Our work highlights how modelling approaches can be a useful complementary method where trials can only be small scale, or where interventions are likely to exert complex effects on wider systems. Future work considering additional causal pathways (including psychosocial pathways) would improve confidence in our findings.

## Declarations

**Ethics approval.**   Model input data are from Understanding Society. The University of Essex Ethics Committee has approved all data collection on the Understanding Society main study and innovation panel waves, including asking consent for all data linkages except to health records. Requesting consent for health record linkage was approved at Wave 1 by the National Research Ethics Service (NRES) Oxfordshire REC A (08/H0604/124), at BHPS Wave 18 by the NRES Royal Free Hospital & Medical School (08/H0720/60) and at Wave 4 by NRES Southampton REC A (11/SC/0274). Approval for the collection of biosocial data by trained nurses in Waves 2 and 3 of the main survey was obtained from the National Research Ethics Service (Understanding Society—UK Household Longitudinal Study: A Biosocial Component, Oxfordshire A REC, Reference: 10/H0604/2). For data used in external validation, ethical approval for each year of the survey was obtained by the Health Survey for England team. No further approval was required for the current analysis of the existing data.

## Supporting information

**S1 Appendix. Short-term impacts of Universal Basic Income on population mental health inequalities in the UK: A microsimulation modelling study.** Table A: Key model assumptions of UKMOD and SimPaths. Table B: Effect estimates for use in Step 2 of SimPaths causal mental health module. Table C: All individual benefits retained and/or suspended in each UBI scenario. Table D: Alternative effect estimates for use in Step 2 of SimPaths causal mental health module during sensitivity analyses. Figure A: Internal validation graphs from the SimPaths GUI contrasting predicted outcomes with observed Understanding Society data from 2011–2017 (yo = years old). Figure B: Cumulative mean prevalence of common mental disorder and poverty by number of model iteratio. Figure C: Prevalence of common mental disorder (CMD) in SimPaths versus the Health Survey for England from 2012–2018. Table E: Population-level economic impacts of Universal Basic Income (UBI) policies modelled in UKMOD. Figure D: Gainers and losers by household income decile (before housing costs) ranging from low to high, with Partial UBI compared with baseline tax/benefit policies in 2023 (Scenario 2). Figure E: Gainers and losers by household income decile (before housing costs) ranging from low to high, with Full UBI compared with baseline tax/benefit policies in 2023 (Scenario 3). Table F: Median income, prevalence of poverty, employment rate, and mean hours worked in baseline scenario and three simulated Universal Basic Income (UBI) scenarios from 2022–2026 (95% uncertainty intervals). Table G: Estimated prevalence of common mental disorders (CMD) and mental health inequalities in baseline scenario and three simulated Universal Basic Income (UBI) scenarios from 2022–2026 (95% uncertainty intervals). Figure G: Estimated relative (left panel) and slope (right panel) indices of inequality by education for common mental disorder (CMD) in modelled Universal Basic Income (UBI) policies from 2022–2026. Table H: Estimated prevalence of common mental disorders (%) in baseline scenario and three simulated Universal Basic Income (UBI) scenarios from 2022–2026 stratified by gender, education, age, and household structure (95% uncertainty intervals. Table I: Structural Sensitivity Analyses—Median income, prevalence of poverty, employment rate, and mean hours worked in baseline scenario and three simulated Universal Basic Income (UBI)

scenarios from 2022–2026 (95% uncertainty intervals). Figure I: Structural Sensitivity Analysis 1, relaxing employment assumptions—Estimated prevalence of common mental disorder (CMD) for modelled Universal Basic Income (UBI) policies from 2022–2026. Figure J: Structural Sensitivity Analysis 2, using economic inactivity effects—Estimated prevalence of common mental disorder (CMD) for modelled Universal Basic Income (UBI) policies from 2022–2026. Table J: Structural Sensitivity Analyses—Estimated prevalence of common mental disorders and mental health inequalities in baseline scenario and three simulated Universal Basic Income (UBI) scenarios from 2022–2026 (95% uncertainty intervals). Figure K: Structural Sensitivity Analysis 1, relaxing employment assumptions—Estimated relative (left panel) and slope (right panel) indices of inequality by education for common mental disorder (CMD) in modelled Universal Basic Income (UBI) policies from 2022–2026. Figure L: Structural Sensitivity Analysis 2, using economic inactivity effects—Estimated relative (left panel) and slope (right panel) indices of inequality by education for common mental disorder (CMD) in modelled Universal Basic Income (UBI) policies from 2022–2026. Figure M: Structural Sensitivity Analysis 1, relaxing employment assumptions—Estimated prevalence of common mental disorder (CMD) for modelled Universal Basic Income (UBI) policies from 2022 to 2026 with 95% uncertainty intervals, stratified by gender (A), education (B), age (C), and household structure (D). Note different scales used for each stratification. Figure N: Structural Sensitivity Analysis 2, using economic inactivity effects—Estimated prevalence of common mental disorder (CMD) for modelled Universal Basic Income (UBI) policies from 2022 to 2026 stratified by gender (A), education (B), age (C), and household structure (D). Note different scales used for each stratification. Table K: Structural Sensitivity Analyses—Estimated prevalence of common mental disorders in baseline scenario and three simulated Universal Basic Income (UBI) scenarios from 2022–2026 stratified by gender, education, age, previous poverty/employment status, and household structure (95% uncertainty intervals). Table L: Analytical Sensitivity Analyses—Median income, prevalence of poverty, and prevalence of unemployment in baseline scenario and three simulated Universal Basic Income (UBI) scenarios from 2022–2026 (95% uncertainty intervals). Figure O: Analytical Sensitivity Analysis, using alternative estimates from systematic reviews—Estimated prevalence of common mental disorder (CMD) for modelled Universal Basic Income (UBI) policies from 2022–2026. Table M: Analytical Sensitivity Analyses—Prevalence of common mental disorders and mental health inequalities in baseline scenario and three simulated Universal Basic Income (UBI) scenarios from 2022–2026 (95% uncertainty intervals). Figure P: Analytical Sensitivity Analysis, using alternative estimates from systematic reviews—Estimated relative (left panel) and slope (right panel) indices of inequality by education for common mental disorder (CMD) in modelled Universal Basic Income (UBI) policies from 2022–2026. Table N: Estimated GHQ Likert score in baseline scenario and three simulated Universal Basic Income (UBI) scenarios from 2022–2026 (95% uncertainty intervals). Table O: Estimated GHQ Likert score in baseline scenario and three simulated Universal Basic Income (UBI) scenarios from 2022–2026 stratified by gender, education, age, previous poverty/employment status, and household structure (95% uncertainty intervals).
(PDF)

## Acknowledgments

We thank Justin van de Ven for his advice and his contributions to the SimPaths codebase, which assisted with our analysis. We also extend our thanks to the project's Advisory Group members for their guidance in shaping this work, and to the broader HEED team for their contributions to the model's development.

This paper is dedicated to the memory of William Old Thomson MBChB FRCS, 1947–2024.

## Author Contributions

**Conceptualization:** Rachel M. Thomson, Anna Pearce, Alastair H. Leyland, S. Vittal Katikireddi.

**Data curation:** Rachel M. Thomson, Daniel Kopasker.

**Formal analysis:** Rachel M. Thomson, Daniel Kopasker.

**Funding acquisition:** Rachel M. Thomson, Anna Pearce, Alastair H. Leyland, S. Vittal Katikireddi.

**Investigation:** Rachel M. Thomson, Anna Pearce, Alastair H. Leyland, S. Vittal Katikireddi.

**Methodology:** Rachel M. Thomson, Patryk Bronka, Matteo Richiardi, Anna Pearce, Alastair H. Leyland, S. Vittal Katikireddi.

**Project administration:** Rachel M. Thomson.

**Software:** Patryk Bronka, Matteo Richiardi, Vladimir Khodygo, Andrew J. Baxter, Erik Igelström.

**Supervision:** Anna Pearce, Alastair H. Leyland, S. Vittal Katikireddi.

**Validation:** Patryk Bronka.

**Visualization:** Rachel M. Thomson, Andrew J. Baxter, Erik Igelström.

**Writing – original draft:** Rachel M. Thomson.

**Writing – review & editing:** Rachel M. Thomson, Daniel Kopasker, Patryk Bronka, Matteo Richiardi, Andrew J. Baxter, Erik Igelström, Anna Pearce, Alastair H. Leyland, S. Vittal Katikireddi.

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
