## [Editor Report · Decision Letter 0]

15 Sep 2023

Dear Dr Thomson, 

Thank you for submitting your manuscript entitled "Short-term impacts of Universal Basic Income on population mental health inequalities in the UK: A microsimulation modelling study" for consideration by PLOS Medicine.

Your manuscript has now been evaluated by the PLOS Medicine editorial staff and I am writing to let you know that we would like to send your submission out for external peer review.

Please re-submit your manuscript within two working days, i.e. by Sep 19 2023 11:59PM.

Kind regards,

Alexandra Schaefer, PhD

Associate Editor

PLOS Medicine

---

## [Decision Letter · Decision Letter 1]

27 Nov 2023

Dear Dr. Thomson,

Thank you very much for submitting your manuscript "Short-term impacts of Universal Basic Income on population mental health inequalities in the UK: A microsimulation modelling study" (PMEDICINE-D-23-02607R1) for consideration at PLOS Medicine. 

Your paper was evaluated by an associate editor and discussed among all the editors here. It was also discussed with an academic editor with relevant expertise, and sent to four independent reviewers, including two statistical reviewers. The reviews are appended at the bottom of this email and any accompanying reviewer attachments can be seen via the link below:

[LINK]

In light of these reviews, I am afraid that we will not be able to accept the manuscript for publication in the journal in its current form, but we would like to consider a revised version that addresses the reviewers' and editors' comments. Obviously we cannot make any decision about publication until we have seen the revised manuscript and your response, and we plan to seek re-review by one or more of the reviewers. 

We expect to receive your revised manuscript by Dec 18 2023 11:59PM. Please email me (aschaefer@plos.org) if you have any questions or concerns.

We look forward to receiving your revised manuscript. 

Sincerely,

Alexandra Schaefer, PhD

PLOS Medicine

plosmedicine.org

GENERAL COMMENTS

Please respond to all editor and reviewer comments.

1) Please cite the reference numbers in square brackets (e.g., “We used the techniques developed by our colleagues [19] to analyze the data”). Citations should be preceding punctuation.

2) Please cite your Supporting Information as outlined here: https://journals.plos.org/plosmedicine/s/supporting-information

3) Please include page numbers and line numbers in the manuscript file. Use continuous line numbers (do not restart the numbering on each page).

4) Please report your study according to the relevant guideline, which can be found here: http://www.equator-network.org/. We suggest including the completed CHEERS (Consolidated Health Economic Evaluation Reporting Standards (CHEERS) 2022 statement) checklist as Supporting Information. When completing the checklist, please use section and paragraph numbers, rather than page numbers. If you do not feel that the checklist is appropriate, please feel free to include an alternative, or if there is not an appropriate option, please leave it out.

5) Please include the Ethics statement in the according section of the online submission form.

6) The terms gender and sex are not interchangeable (as discussed in https://www.who.int/health-topics/gender); please use the appropriate term (including in the supplementary materials) and revise throughout the entire manuscript.

ACADEMIC EDITOR COMMENTS

A very interesting and relevant study. The authors have done a great job at conveying complex methods in an understandable form, although I support reviewer requests for some further explanations to come into the main paper. One reviewer raised an important concern about using a cut-off on the GHQ, which I agree with. All other review comments seemed pertinent and important.

FINANCIAL DISCLOSURE

The funding statement should include: specific grant numbers, initials of authors who received each award, URLs to sponsors’ websites. Also, please state whether any sponsors or funders (other than the named authors) played any role in study design, data collection and analysis, the decision to publish, or preparation of the manuscript. If they had no role in the research, include this sentence: “The funders had no role in study design, data collection and analysis, decision to publish, or preparation of the manuscript.”

COMPETING INTEREST

All authors must declare their relevant competing interests per the PLOS policy, which can be seen here:

https://journals.plos.org/plosmedicine/s/competing-interests

For authors with ties to industry, please indicate whether any of the interests has a financial stake in the results of the current study.

DATA AVAILABILITY STATEMENT

The Data Availability Statement (DAS) requires revision. For each data source used in your study: 

ABSTRACT

1) PLOS Medicine requests that main results are quantified with 95% CIs as well as p values. When a p value is given, please specify the statistical test used to determine it. When reporting p values please report as p<0.001 and where higher as the exact p value p=0.002, for example. For the purposes of transparent data reporting, if not including the aforementioned please clearly state the reasons why not.

2) Throughout, suggest reporting statistical information as follows to improve clarity for the reader “22% (95% CI [13%,28%]; p</=)”. Please amend throughout the abstract and main manuscript. Please note the use of commas to separate upper and lower bounds, as opposed to hyphens as these can be confused with reporting of negative values.

3) Please ensure that all numbers presented in the abstract are present and identical to numbers presented in the main manuscript text.

4) Please include any important dependent variables that are adjusted for in the analyses.

5) Please define ‘UK’ at first use.

6) Please add a unit when discussing age (years, months etc.).

7) Abstract Methods and Findings:

*Please include the number of participants and the length/time frame of follow up/microsimulation model (i.e. what means short-term effect?).

*In the last sentence of the Abstract Methods and Findings section, please describe the main limitation(s) of the study's methodology.

AUTHOR SUMMARY

At this stage, we ask that you include a short, non-technical Author Summary of your research to make findings accessible to a wide audience that includes both scientists and non-scientists. The Author Summary should immediately follow the Abstract in your revised manuscript. This text is subject to editorial change and should be distinct from the scientific abstract. Please see our author guidelines for more information: https://journals.plos.org/plosmedicine/s/revising-your-manuscript#loc-author-summary.

The summary should include 2-3 single sentence, individual bullet points under each of the questions. The last bullet under ‘What Do These Findings Mean?’ point should describe the main limitation of the study's methodology.

It may be helpful to review currently published articles for examples which can be found on our website here https://journals.plos.org/plosmedicine/

INTRODUCTION

1) Please define ‘UK’ at first use.

2) Please expand your introduction about previous research (e.g., what did recent trials show?) and explain the need for and potential importance of your study.

METHODS AND RESULTS

1) Please change “(41.6m)” to “(41.6 million)” or define ‘m’ at first use.

2) Please provide a complete list of model parameters, including clear and precise descriptions of the meaning of each parameter, together with the values or ranges for each, with justification or the primary source cited, and important caveats about the use of these values noted.

3) Please provide a clear statement about how the model was fitted to the data including goodness-of-fit measure, the numerical algorithm used, which parameter varied, constraints imposed on parameter values, and starting conditions.

4) For uncertainty analyses, please state the sources of uncertainties quantified and not quantified [can include parameter, data, and model structure].

5) Please provide sensitivity analyses to identify which parameter values are most important in the model. Uncertainty estimates seek to derive a range of credible results on the basis of an exploration of the range of reasonable parameter values. The choice of method should be presented and justified.

6) Please discuss the scientific rationale for this choice of model structure and identify points where this choice could influence conclusions drawn. Please also describe the strength of the scientific basis underlying the key model assumptions.

7) If you use the points above (2-6), indicate that these are derived from Geoffrey P Garnett, Simon Cousens, Timothy B Hallett, Richard Steketee, Neff Walker. Mathematical models in the evaluation of health programmes. (2011) Lancet DOI:10.1016/S0140-6736(10)61505-X.

8) Under the subheading ‚Sensitivity analysis‘ you write “…two structural sensitivity analyses for Scenarios 3 and 4 (Full UBI).”. Shouldn’t Scenario 4 be ‘Full+ UBI’? Please revise throughout the entire main manuscript. Also, please be consistent/clear in your description of the different scenarios, e.g., Scenario 3, Scenario 3 (Full UBI) or Full UBI. 

9) Please define ‘bn’ at first use.

10) Please define ‘RII’ and ‘SII’ at first use.

11) “Under these conditions, in a reversal of the primary analysis findings, there was a small short-term reduction in prevalence of CMD in Full UBI scenarios: -0.27% (-0.49, -0.05) in 2023, equivalent to a reduction of 112,228 CMD cases (20,783-203,673) (Table 2).” – In this sentence and the following paragraphs, you switch between "Full UBI" and "Full+ UBI", while Table 2 only shows results for the Full+ UBI scenario, please revise.

DISCUSSION

Please discuss what the study adds to existing research and where and why the results may differ from previous research. Please remove the ‘Conclusion’ heading as the one-paragraph conclusion should be part of the discussion.

TABLES

1) Please define abbreviations used in each table (including those in Supporting Information files).

2) Table 1: Please define ‘pensioners’ (i.e. starting age). Please add definitions for the four different scenarios.

3) Table 2: Please define ‘OECD’, ‘UK’, ‘N.’. Please state the meaning of the numbers in brackets/define all numerical values for the reader (e.g., “ABSOLUTE % DIFFERENCE IN CMD PREVALENCE (95% Uncertainty intervals)”).

FIGURES

1) For all Figures, please ensure that you have complied with our figures requirements http://journals.plos.org/plosmedicine/s/figures.

2) Please consider avoiding the use of red and green in order to make your figure more accessible to those with colour blindness. 

3) Please in the figure legend/description, define abbreviations used in each figure (including those in Supporting Information files).

4) Please provide titles, legends and descriptions for all figures (including those in Supporting Information files).

5) Figure 2/Figure 3: Please add definitions for the four different scenarios. Please indicate in the figure caption the meaning of the whiskers. Please show the axis beginning at zero. If this is not possible, please show a break in the axis. Please add x-axis labels. The red line which denotes the reform implementation point should be described as dashed red line (also see the comment #2). For Figure 2, please introduce the abbreviation ‘UBI’ in the figure title (as done e.g., in Figure 3).

6) Figure 3: The color scheme for the different scenarios does not match the one used in Figure 2. Please use a consistent color scheme throughout the manuscript (including the figures in the supplementary materials).

7) Figure 4: Please add definitions for the two different scenarios. Please show the axis beginning at zero. If this is not possible, please show a break in the axis. Please add x-axis labels. The red line which denotes the reform implementation point should be described as dashed red line (also see the comment #2). Please define ‘y’ or write ‘years’ in full. Please mention that the shaded areas show the 95% uncertainty intervals (e.g., “…95% uncertainty intervals (shaded areas)…”).

SUPPLEMENTARY MATERIAL

1) Please define abbreviations used in the supplementary text (at first use), in the supplementary figures and tables.

2) For references in the supplementary material, please see REFERENCES.

3) For supplementary figures and tables, please see the general comments under TABLES and FIGURES (color, abbreviations, titles, descriptions, axis labels, units etc.) and amend accordingly.

4) Figure S7-S16: Please see comments for Figure 2/3/4 and amend accordingly.

5) For supplementary tables: Please ensure to state the unit of the numbers presented and the meaning of the numbers in brackets/define all numerical values (e.g., for Table S8 and S11 change to “Estimated prevalence of common mental disorders (%, 95% Uncertainty intervals)…”). 

REFERENCES

1) PLOS uses the numbered citation (citation-sequence) method and first six authors, et al.

2) Please ensure that journal name abbreviations match those found in the National Center for Biotechnology Information (NCBI) databases (http://www.ncbi.nlm.nih.gov/nlmcatalog/journals), and are appropriately formatted and capitalised.

3) Where website addresses are cited, please specify the date of access. 

4) Please also see https://journals.plos.org/plosmedicine/s/submission-guidelines#loc-references for further details on reference formatting. 

Comments from the reviewers:

Reviewer #1: This is a well-written paper on an interesting issue of public health and policy salience - UBI is currently of great political interest and the paper makes a contribution to this discourse. As the authors point out, in the absence of rigorous trials of UBI, microsimulation is a valuable tool for exploring the range of potential outcomes that might follow the introduction of a UBI. But, of course, any microsimulation rests heavily on the assumptions being made and this study, although articulating and discussing these limitations, is perhaps less forthright about the resultant very conservative nature of their findings than it might be.

The study is well situated in the context of existing literature, although - perhaps because of the timing of submission - recent UK based microsimulation studies of basic income are not referenced:

* Parra-Mujica, F., Johnson, E., Reed, H., Cookson, R. & Johnson, M. (2023) Understanding the relationship between income and mental health among 16- to 24-year-olds: Analysis of 10 waves (2009-2020) of Understanding Society to enable modelling of income interventions A. Moretti (ed.). PLOS ONE. 18(2): e0279845. DOI: 10.1371/journal.pone.0279845

* Reed, H.R., Johnson, M.T., Lansley, S., Johnson, E.A., Stark, G. & Pickett, K.E. (2023) Universal Basic Income is affordable and feasible: evidence from UK economic microsimulation modelling. Journal of Poverty and Social Justice. 31(1): 146-162. DOI: 10.1332/175982721X16702368352393.). 

More attention could also be given to the different pathways through which a UBI might be expected to influence mental health (see, for example, the models of UBI impact in Johnson, M.T., Johnson, E.A., Nettle, D. & Pickett, K.E. (2022) Designing trials of Universal Basic Income for health impact: identifying interdisciplinary questions to address. Journal of Public Health. 44(2): 408-416. DOI: 10.1093/pubmed/fdaa255 and Huss R. Can universal basic income reduce poverty and improve children's health?Archives of Disease in Childhood Published Online First: 30 March 2023. doi: 10.1136/archdischild-2022-324799). These models theorize that the ways in which UBI affects health go far beyond the material effects of income alone and need to be viewed within a broader social determinants of health framework.

The methods of the study are well-described. This is a useful case study application of the causally informed mental health module within HEED. The sensitivity analyses related to potential impacts on employment are a strength of the modelling in light of the findings related to employment in the Finnish UBI pilot. 

The Discussion section does include some discussion of the two major assumptions of the modelling - that macroeconomic impacts of UBI have no effect on mental health and that income changes due to UBI will have the same mental health impact as other changes in income. These are clearly quite important assumptions, given relationships between income inequality and mental health and indeed the modelled effect on the Gini coefficient (see, for example, Ribeiro, Wagner Silva, et al. "Income inequality and mental illness-related morbidity and resilience: a systematic review and meta-analysis." The Lancet Psychiatry 4.7 (2017): 554-562 and Patel, V., Burns, J.K., Dhingra, M., Tarver, L., Kohrt, B.A. and Lund, C. (2018), Income inequality and depression: a systematic review and meta-analysis of the association and a scoping review of mechanisms. World Psychiatry, 17: 76-89. https://doi.org/10.1002/wps.20492) and the likely importance of income stability, increased levels of societal trust and other psycho-social benefits of a true UBI (as per models referenced above). The authors do recognise that their estimates are likely to be conservative, but perhaps don't give the weight to these assumptions in their inference that they might merit. A useful next step would be to incorporate at least the impact on income inequality into future modelling.

Reviewer #2: This paper is a microsimulation model of the effect of a universal basic income (UBI) on mental health. This is a relevant topic for PLOS Medicine as there is strong evidence for impact of income on mental health, but limited evidence for the effect of structural interventions that target income on mental health (or other population health outcomes). The methodological approach is appropriate given feasibility issues of implementing a UBI at the population-level, and this paper will complement results from smaller pilots globally that are targeted based on income or other population characteristics. Potential issues with the extent to which these scenarios may be grounded in practical application are addressed through the input of UBI leaders and health and government decision-makers in selecting the parameters of the four UBI scenarios. The assumptions of the model are important for microsimulation to make contributions in this field of research, and while the authors raise the issue of macroeconomic impacts, these were not included in sensitivity analysis. This is the paper's major weakness and requires either inclusion in the microsimulation or more justification as to why this was not included. I recommend major revisions before this paper can be published with PLOS Medicine. 

Major revisions

Methods:

-The authors identify several structural and theoretical assumptions of the model, including that any macroeconomic impacts of the policy intervention would not affect the outcomes. This is a significant weakness of the model that requires further justification and discussion, or to be addressed through an additional sensitivity analysis. It is reasonable to expect that mental health services and supports would be less available with the introduction of a UBI either owing to cut-backs related to UBI financing, or labour markets impacts of people leaving lower-paid jobs social care roles and making care less available. While this may not have impacts on people with more transitory or situational mental health concerns related to work or income stress that may be addressed through the UBI, it could worsen mental health for those who require more fulsome supports. 

Minor revisions

Introduction 

-The authors indicate that that mental health has been worsening over the past 15 years "against a background of economic crises, austerity policies, and COVID-19", but only one citation shows some empirical evidence for this relationship. Suggest providing more explanation of other trends that are attributable to the increase in mental health disorders (i.e., generational shift in mental health disorders among adolescents and young people, increased use of digital media, etc). It may otherwise be interpreted that greater economic security should result in a more significant decrease in mental disorders than was found through the microsimulation model. 

-The authors use the definition of a UBI (versus a guaranteed income or basic income) which includes universality, however the examples provided (Finland, Ontario, and individuals leaving the care system) do not meet that criteria. Revise to indicate that these pilots are not a UBI, which would be consistent with the authors statement that "no fully universal UBI has been trialed". 

-The statement that no fully universal UBI has been trialed is accurate for high-income countries, but some low-income country trials do meet the criteria of universality in smaller geographic areas (e.g., GiveDirectly in rural Kenya). Revise to indicate that no fully universal UBI has been trialed in a high-income context. 

Methods

-The authors indicate that the mental health module was developed using empirical epidemiological estimates of the effects of economic transitions on mental health - where were estimates derived from? 

Results: 

-The authors stratify results based on several sociodemographic factors (gender, education, age, and household structure) but there is no stratification based on level of taxation. Under the partial, full and full+ UBI scenarios, there is a significant change in taxation for those who are currently 'middle income', e.g., 70% from 30K and 85% from 50K for the full and full+ UBI (unless I have misunderstood the income tax rates in Table 1, in which case more explanation is required). I would expect some increase in CMDs is from people who were previously in a higher income bracket but whose income is now lower post-taxation, and so it would be interesting to see if the increases are higher in that group. 

Discussion:

-The authors state that "there is little evidence UBI-like interventions are associated with large increases in unemployment", however the evidence for minimal impacts is thin. Hum (1993) was published before current labour market trends (e.g., precarity, frictional employment) that could impact the behavioural response to a UBI, and de Paz-Banez (2020) indicates significant methodological weaknesses in what is used as proxies for a UBI While there is still relatively good consensus that UBI would not impact unemployment, suggest citing more robust literature given how important increases in unemployment were to the sensitivity analysis. I suggest the authors draw from more recent literature ( see Marinescu, Skandalis and Zhao; Scott, Altonji et al.) which concluded that COVID-19 benefits resulted in minimal work disincentives from the expansion of benefits during COVID-19. Although these benefits were temporary, they may be of more relevance to the behavioural response to a UBI than the older literature that is cited.

Reviewer #3: Thank you to the authors for submitting this paper, I very much enjoyed reading it. I definitely think it has the potential for publication in this journal, but I have a number of comments that I think should be addressed or considered before publication. I have listed them in roughly the order they appear in the text and have indicated which comments I think are minor and which are major. I hope the comments are useful and potentially improve the paper. 

Major comment - I think a significant amount of further justification needs to given regarding the choice to dichotomise the GHQ-12 and use this as your primary outcome measure. I completely understand that it is easier to interpret (i.e. you can estimate an approximate cost per common mental disorder avoided), but to me it feels as though you are just throwing away potentially useful information. Although the interpretation of the results would clearly be a little bit more difficult, I think looking of the relationship in a little bit more granular detail would be useful rather than a dummy variable. At the very least I think these results should be presented as part of the online appendix. 

Major comment - I fully appreciate that it is a very complicated (and impressive) set of methods, but I really think that further detail in the main text is needed regarding the Simpaths model, especially the transitions between the different states (for example the probabilities and distributions used). I know some of this information is included in the online appendices, but I think it is really needed in the main text in order for the paper to flow. The model comes across as a bit of a black box at the moment.

Major comment - I would also like some discussion of the 'causal' methods used in the main text, including the g-computation methods used. They are briefly referenced in the appendix (and weirdly the acknowledgements) but for me this isn't enough. They are an important aspect of the paper so should be in the main text somewhere. This is contributing to the 'black box' feel of the model as it currently reads. I would move Table S2 (and potentially Table S4) from the appendix into the main text. 

Major comment - The measures of inequality used in certain parts of your analyses (relative index of inequality, slope index of inequality, gini coefficient) need to be explained in the methods. A reader with no background quantitative measures of inequality such as these would have no idea what they mean. 

Major comment - I think you need to further emphasise the significant level of uncertainty present in the vast majority of the results you present. Looking at Table 2, it looks like your "Main Analysis" in 2023 is the only estimate where the confidence intervals do not contain 0. Can you comment further on the possible reasons for this significant level of uncertainty? Where in the model do you think it is coming from?

Minor comment - Is the Full UBI (orange) line missing from the left hand panel of Figure 2?

Minor comment - I would use the same scale for each of the four panels in Figure 4 so it is easier for the reader to compare the different sub-groups. I also personally find the colour scheme difficult to read - that might just be me though!

Minor Comment - In the sensitivity analysis section you note that you "modified the utility values... so employment rates remained constant". I think you need to explain this comment further, what utility values are you talking about? It isn't clear

Minor Comment - Is there any particular reason you used the median values in the stochastic uncertainty analysis? I personally have only ever seen the mean values used in these instances. One sentence explaining why you use the median values will probably do.

Minor comment - Similar to a previous comment, but in the discussion can you comment on the possible reasons why the impacts on mental health are relatively short lived and don't sustain over the long term?

Minor comment - Does "your recent systematic review" reference much of the economics literature which has used IV/RDD methods to identify a causal impact of income on health? If so I think it would be briefly worth mentioning here the difficulties of using these methods and the drawbacks in terms of interpretation of the results from such studies. 

Minor comment - Are you aware of any applied economic literature which has looked at the willingness to pay of avoiding mental illness/disorders? If there are any papers regarding this it would be good to bring into your discussion to put your "£600,000 per case of CMD avoided" into some context.

Reviewer #4: This is a really interesting and well written paper on the potential impact of implementing a universal basic income on mental health in the UK. 

I would like to commend the authors on their inclusion of PPIE in this study and the role they played in shaping the research and also on moving the focus away from fiscal neutrality. However, I would have liked to have seen fiscal neutrality as a scenario as they is likely needed from a political perspective.

I thought the choice of timeframe (5 years) and the introduction on UBI one year into the cycle will help with how the implementation of such a policy would work in reality in the UK. However, would it have been worthwhile looking at the implementation of the partial UBI and then perhaps a transition to a full UBI within the same model run but with a longer time horizon to allow this progression? I believe that any government would more likely follow this route rather than introduce full or full+ straight off. Furthermore, from a policy perspective, I would have also liked to have seen the reverse i.e. to model the successor government post-five years removing the policy and then seeing how quickly the impact is reversed.

Minor point - I would like to see some justification of the use of 1000 simulation runs for each analysis in your uncertainty analysis.

Minor point - table 2: I am not sure anyone finds values of -0.00 helpful - would it be possible to change to > unless the value is actually 0. 

I would also like to see reference to the review by Pinto et al on Exploring different methods to evaluate the impact of basic income interventions: a systematic review in your background prior to introducing the microsimulation approach employed in your paper.

[LINK]

---

## [Decision Letter · Decision Letter 2]

19 Jan 2024

Dear Dr. Thomson,

Thank you very much for re-submitting your manuscript "Short-term impacts of Universal Basic Income on population mental health inequalities in the UK: A microsimulation modelling study" (PMEDICINE-D-23-02607R2) for review by PLOS Medicine.

I appreciate your detailed response to the editors' and reviewers' comments. I have discussed the paper with my colleagues and the academic editor, and it has also been seen again by all three original reviewers. The changes made to the paper were satisfactory to the reviewers. As such, we intend to accept the paper for publication, pending your attention to the editorial comments below in a further revision. When submitting your revised paper, please once again include a detailed point-by-point response to the editorial comments.

[LINK]

In revising the manuscript for further consideration here, please ensure you address the specific points made by each reviewer and the editors. In your rebuttal letter you should indicate your response to the reviewers' and editors' comments and the changes you have made in the manuscript. Please submit a clean version of the paper as the main article file. A version with changes marked must also be uploaded as a marked up manuscript file. Please also check the guidelines for revised papers at http://journals.plos.org/plosmedicine/s/revising-your-manuscript for any that apply to your paper. 

We ask that you submit your revision within 1 week (Jan 26 2024). However, if this deadline is not feasible, please contact me by email, and we can discuss a suitable alternative.

Don’t hesitate to contact me directly with any questions (aschaefer@plos.org). If you reply directly to this message, please be sure to ‘Reply All’ so your message comes directly to my inbox.  

We look forward to receiving the revised manuscript.

Sincerely,

Alexandra Schaefer, PhD

Associate Editor 

PLOS Medicine

plosmedicine.org

Requests from Editors:

ABSTRACT

1) Please note that you can remove the units from the 95% UI values in parentheses. Since the corresponding unit is presented with the foregoing value, it is not necessary to repeat the units. Also, please ensure to add 95% UI before each set of parentheses and that the 95% UI values are presented in parentheses (please see comment 3) for an example).

2) ll.28-29: Please add "to our knowledge" or something similar.

3) ll.45-46: It seems the second pair of parentheses should directly follow ’74.1%’. Please revise. Editorial suggestion: “(for Full+ from 78.9% (95% UI [77.9,79.9]) to 74.1% (95% UI [72.6, 75.4]))”

4) In the last sentence of the Abstract Methods and Findings section, please describe the main limitation(s) of the study's methodology.

AUTHOR SUMMARY

We suggest changing the last bullet under ‘What Do These Findings Mean?’ point to: The main limitation of our modelling study is that it looks at how UBI would influence mental health only through income and employment, while other pathways were not considered in the analysis.

INTRODUCTION

ll. 110-111/115-116: Please add “to our knowledge” or similar.

METHODS AND RESULTS

1) l.248. Please define ‘OECD’ at first use.

2) l.329 and ongoing: Please see comment 1) under ABSTRACT and adjust the statistical reporting accordingly (for example, line 329: 0.01% (95% UI [0.00, 0.03])).

3) ll.331-333: Please provide reference to the relevant graph and/or table.

4) ll.336-337: Please check whether Table S6, Appendix is the (only) appropriate reference here. It seems the relevant data of employment rates described here are also visible in Figure 2 (right panel).

5) ll.337-338: Please provide reference to the relevant graph and/or table.

6) ll.338-340: You state that the employment effects of the Full UBI policies were sustained throughout the study period after implementation. However, when looking at Figure 2, one can see that while employment rates initially fell in both Full UBI scenarios after policy implementation, they rose steadily in the following years. You might consider modifying this conclusion accordingly.

7) ll.443-450: We suggest briefly reiterating what the GHQ Likert scores reflect, as it may be easier for readers who are not as familiar with the content to follow this section.

FIGURES

1) Please provide a figure description for Figure 1.

2) Figure 4: In the figure description, we suggest adding a definition of low, medium and high education.

REFERENCES

1) Please thoroughly revise all references and ensure that journal name abbreviations match those found in the National Center for Biotechnology Information (NCBI) databases (http://www.ncbi.nlm.nih.gov/nlmcatalog/journals), and are appropriately formatted and capitalised (e.g., for reference [1] The Lancet Regional Health – Europe should be Lancet Reg Health Eur)

2) When specifying the date of access, please write “accessed” instead of “cited”.

3) Please revise the format of reference [65].

SUPPLEMENTARY MATERIAL

1) p.7, please change to: “…considerable debate on the ‘best’ way to fund UBI.”

2) Figure S1/S13/S14 (and where applicable): Similar to Figure 4, we suggest adding a definition of low, medium and high education.

3) Table S4: Please change "Full years UBI only" to "Full UBI years only".

4) Table S8/S11/S15: We suggest adding a definition of low, medium and high education and changing “Younger 25-44y” and “Older 45-64y” to “Age group: 25-44 years” and “Age group: 45-64 years”.

5) Table S6/S9: We wondered why you chose to present only the differences and not the actual values for average weekly hours worked, and why you chose not to present the differences for median annual income, poverty, and employment.

6) Figure S12: Please note that in Figure S12 you use the term " economic inactivity causal effects" while in others you use the term " economic inactivity effects" - please try to be consistent.

7) Please revise the references according to the comments under REFERENCES. 

SOCIAL MEDIA

To help us extend the reach of your research, please provide any X (formerly known as Twitter) handle(s) that would be appropriate to tag, including your own, your coauthors’, your institution, funder, or lab. Please respond to this email with any handles you wish to be included when we tweet this paper.

Comments from Reviewers:

Reviewer #1: The authors have responded in full to my points, and to the points raised by the other reviewers. The responses are detailed and thoughtful and I believe the paper will now make a significant and useful contribution to the literature.

Reviewer #2: Response to major comments: The authors have provided a detailed and comprehensive response to the major suggested revisions, specifically, the concerns raised by several reviewers on the macroeconomic impacts of the UBI on the outcomes of interest. The paper is now much clearer on the limitations posed by the key assumptions, and the justification of the considerable additional complexity of integrating a macroeconomic model at this stage are reasonable.

Response to minor comments:

R2.7: I appreciate the authors' response that stratification of results based on tax brackets is not possible due to the separation between the UBI modelling within UKMOD to the dynamic health modelling within SimPaths. While the UKMOD output includes a calculation of the Gini coefficient for each policy scenario, it may be interesting to include a comparison between current income distribution (by tax brackets) and income distribution under each of the scenarios as an appendix (in addition to Figure 2), with a very brief note in the discussion that references the current distribution of CMDs across the income distribution, and a note that the minimal change under the UBI scenarios may be due to changes in the proportions of people in each tax bracket.

Reviewer #3: Thank you to authors for their detailed response to each of my comments. I am now happy to approve this paper for publication.

[LINK]

General Editorial Requests

To submit your revised manuscript:

---

## [Editor Report · Decision Letter 3]

5 Feb 2024

Dear Dr Thomson, 

On behalf of my colleagues and the Academic Editor, Charlotte Hanlon, I am pleased to inform you that we have agreed to publish your manuscript "Short-term impacts of Universal Basic Income on population mental health inequalities in the UK: A microsimulation modelling study" (PMEDICINE-D-23-02607R3) in PLOS Medicine.

I appreciate your thorough responses to the reviewers' and editors' comments throughout the editorial process. We look forward to publishing your manuscript, and editorially there are only two remaining minor stylistic/presentation points that should be addressed prior to publication. We will carefully check whether the changes have been made. If you have any questions or concerns regarding these final requests, please feel free to contact me at aschaefer@plos.org.

Please see below the minor points that we request you respond to:

1) l.345: Please change to: "75.69% (95% UI 74.44, 76.92)"

2) In the references (including those in the Supporting Information), please note that the date of access to the website should include the day, month, and year. Please revise accordingly.

PRESS

Sincerely, 

Alexandra Schaefer, PhD 

Associate Editor 

PLOS Medicine